# Weather Patterns Associated with DON Levels in Norwegian Spring Oat Grain: A Functional Data Approach

**DOI:** 10.3390/plants11010073

**Published:** 2021-12-27

**Authors:** Anne-Grete Roer Hjelkrem, Heidi Udnes Aamot, Morten Lillemo, Espen Sannes Sørensen, Guro Brodal, Aina Lundon Russenes, Simon G. Edwards, Ingerd Skow Hofgaard

**Affiliations:** 1Division of Food Production and Society, Norwegian Institute of Bioeconomy Research (NIBIO), 1431 Ås, Norway; aina.lundon@nibio.no; 2Division of Biotechnology and Plant Health, Norwegian Institute of Bioeconomy Research (NIBIO), 1431 Ås, Norway; heidi.udnes.aamot@nibio.no (H.U.A.); guro.brodal@nibio.no (G.B.); ingerd.hofgaard@nibio.no (I.S.H.); 3Department of Plant Sciences, Norwegian University of Life Sciences (NMBU), 1432 Ås, Norway; morten.lillemo@nmbu.no; 4Graminor Breeding, 2322 Ridabu, Norway; espen.sorensen@graminor.no; 5Centre for Integrated Pest Management, Harper Adams University, Newport TF 10 8NB, UK; sedwards@harper-adams.ac.uk

**Keywords:** deoxynivalenol, Fusarium, FHB, mycotoxin, phenological model

## Abstract

*Fusarium graminearum* is regarded as the main deoxynivalenol (DON) producer in Norwegian oats, and high levels of DON are occasionally recorded in oat grains. Weather conditions in the period around flowering are reported to have a high impact on the development of Fusarium head blight (FHB) and DON in cereal grains. Thus, it would be advantageous if the risk of DON contamination of oat grains could be predicted based on weather data. We conducted a functional data analysis of weather-based time series data linked to DON content in order to identify weather patterns associated with increased DON levels. Since flowering date was not recorded in our dataset, a mathematical model was developed to predict phenological growth stages in Norwegian spring oats. Through functional data analysis, weather patterns associated with DON content in the harvested grain were revealed mainly from about three weeks pre-flowering onwards. Oat fields with elevated DON levels generally had warmer weather around sowing, and lower temperatures and higher relative humidity or rain prior to flowering onwards, compared to fields with low DON levels. Our results are in line with results from similar studies presented for FHB epidemics in wheat. Functional data analysis was found to be a useful tool to reveal weather patterns of importance for DON development in oats.

## 1. Introduction

Occasionally high levels of mycotoxins are recorded in Norwegian oat grains [1,2]. Despite the effort put into breeding for increased resistance to *Fusarium graminearum*, an important member of the Fusarium Head Blight (FHB) disease complex and regarded as the main producer of the mycotoxin deoxynivalenol (DON) in Norwegian oats [1], the varieties are still only moderately resistant [3]. In years with high risk of DON, it would therefore be advantageous if the DON-risk could be predicted, as part of a decision support system (DSS) for farmers so that a possible fungicide treatment to control *F. graminearum* infection could be applied at flowering when infection risk is high. For the grain industry, a prediction of the quality of the year’s harvest related to DON content would additionally give an indication of which areas are potentially at risk and thus where extra effort should be put into DON analysis of grain lots.

FHB epidemics in wheat are highly associated with weather conditions in the growing season, thus weather conditions are the main drivers of statistical and process based models developed to estimate the risk level for Fusarium infection and mycotoxin development [4,5,6,7,8,9,10,11,12,13,14]. The majority of the models that are published concern the association between weather conditions around the period of flowering in wheat and the development of *F. graminearum* and DON in wheat grains. Weather conditions impact the growth and development of both host plants, fungal pathogens as well as the degree of infection and disease development [6,15,16]. Thus, weather patterns associated with a risk of Fusarium infection and DON development may differ throughout the growth season. To develop models with adequate accuracy, it is therefore crucial to identify important weather patterns and the time periods of special importance for the development of FHB and DON. Window-pane methodology has been used for this purpose [17], identifying weather variables during different overlapping time periods (windows) with high correlation to FHB in wheat. In Hjelkrem et al. [8], a similar methodology was used for DON development in oats, but instead of overlapping fixed-length windows, time periods related to the host phenological stage were used. Both methods discretize the continuous weather time series into fixed length windows of different starting points and lengths to identify important weather factors relative to the risk of DON in harvested oats. To assess the continuous weather time series, Shah et al. [18] proposed the application of functional data analysis and then performed a study to identify weather patterns linked to FHB epidemics in wheat. For several weather conditions, distinct smoothing curves for epidemics versus non-epidemics were developed from 120 days prior to flowering. Differentiation of FHB epidemics from non-epidemics were especially observed for daily average moisture related conditions and the largest differences were seen close to flowering. The authors did not include rainfall in their analysis as they considered this variable to be more site-specific than the resolution of their input data. Some of the observations corresponded with previous findings on the relation between weather variables and the development of *F. graminearum* [19,20]. Shah et al. [18] suggested that predictive weather-based modelling of FHB would benefit from the use of several input variables over potentially different time frames relative to flowering, and they further used this approach to develop functional models which overall fitted the data better than previously described standard logistic regression models for prediction of FHB in wheat [13]. The authors observed that weather summaries encompassing the time point of flowering were better input predictors than summaries that were restricted to either side of flowering.

The relationship between weather conditions and the accumulation of DON in oats is not well explored [8,21], and no prediction model has yet been found that is suitable to be launched on the Norwegian DSS system VIPS (Varsling Innen PlanteSkadegjørere, http://www.vips-landbruk.no/, access date: 17 November 2021) developed for assisting Norwegian farmers and advisors in decisions on whether or when to apply a pesticide.

Field operations are often linked to a certain crop growth stage, and mathematical models for the prediction of growth stages in cereals can thus be useful for farmers to facilitate the planning of such operations. Also, as the disease susceptibility of a plant may differ with growth stage, growth stage models can be included as important parts of plant disease prediction models [8]. The growth and development of oats is closely associated with air temperature and photoperiod. Several simple mathematical models for the prediction of phenological growth stages in oats exist [22,23,24,25] including one model (OPM1.0) for Norwegian varieties and conditions [8]. The Norwegian OPM1.0 model was mainly based on observations of oat development during the heading/flowering and dough stages. Thus, we anticipate that the model would be considerably improved with a new structure or at least one that is more finely calibrated, if field observations at earlier growth stages were included. In order to establish prediction models for DON levels in oat grains, more information on the relationship between weather conditions at specific developmental stages of oats and DON risk is needed.

In this publication we present (i): an updated model for the prediction of phenological growth stages in spring oats based on weather data and (ii): weather patterns associated with DON levels in oats identified by plotting functional mean curves for oat grain lots with high versus low DON content.

## 2. Results

### 2.1. Development of an Improved Model for Prediction of Phenological Growth Stages of Oats

A number of oat phenology models (Appendix A) were developed based on observations of the variety Belinda. According to the cross site-year validation technique, Bayesian calibration was conducted by running two Markov chains in parallel for 150,000 iterations. For each model, the root mean square error of validation (RMSEV) was calculated using the maximum a posteriori parameter estimates (MAP) and given as a mean value (and standard deviation) over the splits in the cross site-year validation.

The model (Appendix A) fit to the validation data from the different varieties is given in Table 1. Most models seemed to fit similarly well to the validation data from variety Belinda, with the error term barely above four as a mean value over the splits in the site-year validation. Overall, Model 18 (four parameters and two environmental variables) was the best model with lowest RMSEV (4.1), closely followed by Model 1 (two parameters and one environmental variable), Model 2 (three parameters and one environmental variable) and Model 17 (three parameters and two environmental variables) with a slightly higher RMSEV (4.2). The difference in fit between these four models was very low, whereas the complexity differed highly. The simplest model (Model 1) was therefore selected as the overall best model to estimate the phenological growth stages of oat variety Belinda. Four models stood out with a much higher error term (RMSEV between 5.2 and 6.5), which encompassed all models with a base temperature of 5 °C (Model 3, Model 11 and Model 16) and one model with a base temperature of 0 °C before the middle of heading and 9 °C thereafter (Model 5).

By considering the standard deviation (std) of the error term (Table 1), instead of the mean value for the variety Belinda, the difference in model fit across the different splits in the cross site-year validation is determined. The standard deviation varies between 0.9 and 3.6, which reflects a relatively high variation. The individual RMSEV for each split is given in the Appendix A. Generally (except two models), Apelsvoll 2017 had the lowest RMSEV (2.5 to 4.5) for the models evaluated, whereas Apelsvoll 2019 had the highest RMSEV (4.4 to 7.0) (except four models) (followed by Staur 2019). For the four models with the lowest error terms, RMSEV varied between 2.9 and 5.8 (std of 1.1) for Model 18, whereas for Model 1, Model 2 and Model 17, it varied between 2.8 and 5.9 (std of 1.0) when considering the variety Belinda, only.

The four models with RMSEV below or equal to 4.2 for Belinda (Model 1, Model 2, Model 17 and Model 18), also had low RMSEV for the varieties Ringsaker, Haga, Odal, Vinger and Våler when regarded separately, and for all 13 varieties when merged together (Table 1). The variation in RMSEV was much smaller between varieties than between splits (site-year combinations). For the late maturing varieties Odal and Våler and the early maturing variety Haga, these models had RMSEV values in the same range (4.1 to 4.2) as Belinda (late maturing variety). For the two other varieties Vinger (late maturity) and Ringsaker (early maturity), and for all varieties merged, the RMSEV was slightly higher (4.3 to 4.4). Consequently, there were no clear differences in model fit between early and late maturing varieties. For Model 1 and Model 17, the variation in RMSEV between varieties was smaller (4.1 to 4.2) than for Model 2 and Model 18 (4.1 to 4.4).

Model 1 is selected as the overall best model, due to its good fit to validation data from all the included varieties, its simplicity with only two free parameters and one environmental variable, small standard deviation between splits in the cross site-year validation and small variation in RMSEV between varieties. The estimated MAP parameter values were *α* = 4.0561 and *β* = 0.0024. This model is the same model as the oat phenology model (OPM1.0), derived in Hjelkrem et al. [8], but with new updated parameter values. This model (Model 1) is implemented and available on the Norwegian DSS system VIPS (http://www.vips-landbruk.no/, access date: 17 November 2021) and it is hereafter named OPM1.1.

In Figure 1a, observed phenological growth stages are plotted against the number of days from sowing in the six varieties. In Figure 1b, data on all the observed phenological growth stages are plotted against degree days for the same varieties together with the output data from OPM1.0 [8] and OPM1.1 (Model 1, derived in this study). The output from OPM1.0 and OPM1.1 mainly differs in the early phenological stages of oats, while this difference is gradually reduced with increasing degree days until BBCH 85 from which the models have a similar output. Appendix A, presents output from the two models together with the independent validation data for each of the splits in the cross site-year validation separately, and Appendix A gives the training data of Belinda together with the validation data of Belinda.

### 2.2. Functional Data Analysis of Weather Conditions Related to DON in Oat Grain

In order to identify weather conditions of importance for the development of DON in oats, functional data analysis with smoothing functions was performed on time series of weather-based variables connected to a total of 201 oat fields, including 121 fields with low DON levels (<500 µg DON per kg harvested oats), and 80 fields with high DON levels (≥500 µg/kg). Possible associations between the average daily observations of selected weather variables were also examined for the time period from 90 days prior to 30 days posterior to estimated flowering. A significant negative correlation (*r* = −0.62, *p* < 0.01) was observed between the daily difference in minimum and maximum temperatures (T.MINMAXDIFF; °C) and the average daily relative humidity (RH.A; %) in our dataset (Appendix A). Furthermore, a significant positive correlation (*r* = 0.57, *p* < 0.01) was observed between the average daily vapor pressure deficit (VPD.A; kPa) and the average daily temperature (T.A; °C) (Appendix A), while a significant negative correlation (*r* = −0.84, *p* < 0.01) was observed between RH.A and VPD.A (Appendix A). When studying the accumulated number of hours for the time period from 90 days prior to 30 days posterior to estimated flowering, a significant positive correlation (*r* = 0.74, *p* < 0.01) was observed between vapor pressure deficit lower than 0.635 kPa (VPD.L0635.CHD) versus relative humidity greater than 75% combined with temperature between 9 and 26 °C (TRH.9T26nRHG75.CHD, Appendix A). The correlation increased (*r* = 0.91, *p* < 0.01) when assessing a narrower time interval around estimated flowering (30 days prior to 30 days posterior to estimated flowering).

The day of mid flowering (BBCH 65) was estimated for each field using the phenological model (OPM1.1), and cubic B-splines were further used to develop a smoothing function from 90 days prior to 30 days posterior to estimated flowering. The number of days from sowing to estimated flowering differed between 51 to 89 days for all the fields included (see Figure 2 where dash-dotted lines indicate the start and end of the sowing period across all fields), which was also the case for fields with low DON levels. A narrower time interval was detected for fields with high DON levels (between 59 and 79). Still, within each group of fields with either low or high DON levels, the time period between sowing and estimated flowering differed between 60 and 80 days for the majority of the fields (>92%). The time period from estimated flowering to harvest differed from 25 to 97 days across all fields with a mean of 50 days. This means that weather observations 25 days after estimated flowering and onwards are not relevant for all the fields included. However, for the majority (98%) of the fields within each group of fields with either low or high DON levels, the time period from estimated flowering to harvest was between 30 and 70 days. Significant differences between the functional mean curves for oat fields with high versus low DON levels are shown as asterisk in the figures (5% level).

In Figure 2 and Figure 3, smoothed functional means are plotted together with daily outputs of weather-based variables for each oat field in the time period from 90 days prior to 30 days posterior to estimated flowering, with separate panels for the oat fields with high DON levels (left column of panels, Figure 2a,d,g,j,m and Figure 3a,d,g,j,m) and with low DON levels (mid column of panels, Figure 2b,e,h,k,n and Figure 3b,e,h,k,n) for different weather variables. The remaining panels (right column of panels, Figure 2c,f,i,l,o and Figure 3c,f,i,l,o) indicate the difference between the functional means for high versus low DON levels, and with asterisks at the x-axis indicating periods with significant differences between these two groups at a 5% level. In the Appendix A, the cumulative weather variables are presented for the time period from 30 days prior to 30 days posterior to estimated flowering (Appendix A) and as a function of the estimated phenological growth stages (Appendix A). The latter two figures present weather conditions observed during the plant’s growth period only, which may be from 51 to 89 days prior until 25 to 97 days posterior to estimated flowering for the various fields.

The lines in Figure 2a,b indicating the functional means for daily average temperatures (T.A) related to the number of days from estimated flowering, had curved shapes with maximum levels around estimated flowering. Oat fields with high levels of DON had a generally higher daily average temperature in the period around sowing and consistently lower daily average temperature from the period around estimated flowering onwards compared to oat fields with low levels of DON (Figure 2c). In a five-day period from sowing, the mean daily temperatures for the oat fields with high levels of DON were almost 1 °C higher than the mean temperature for the oat fields with low levels of DON, 11.0 and 10.1 °C respectively. Contrary to this, in a five-day period around estimated flowering (BBCH 65), the mean daily air temperatures for the oat fields with high levels of DON were more than 1 °C lower than the mean temperature for the oat field with low levels of DON, 15.1 and 16.3 °C, respectively.

The cumulated number of hours with temperature lower than 9 °C (T.L9.CHD) are presented as functional means in Figure 2d,e, with less variation in the records for the fields with high DON levels compared to the records for the fields with low DON levels. The separation between the curves (Figure 2f) shows a consistently and significant lower mean functional curve for the fields with high DON contamination compared to the curve for the group with low DON contamination. The rate of accumulation differed especially between fields with high versus low DON levels in the period around sowing. On the contrary, when starting the accumulation 30 days prior to estimated flowering (Appendix A), a consistently higher mean functional curve was observed for the fields with high DON contamination compared to the curve for the group with low DON contamination. However, the difference was only significant from 25 days posterior to estimated flowering and onwards. When studying the association between T.L9.CHD at estimated growth stages and DON levels, consistently but not significantly lower values were detected in fields with high compared to low DON levels (Appendix A).

The functional means for the cumulative number of hours with temperature between 10 °C and 15 °C (T.10T15.CHD) were calculated relative to the number of days from estimated flowering (Figure 2g,h). The separation between the curves showed a consistently higher value of the functional mean for fields with high DON levels, and this difference was significant for the entire period (Figure 2i). The rate of accumulation differed especially between fields with high versus low DON levels in the period around sowing and from 10 days prior to estimated flowering and onwards. When starting the accumulation 30 days prior to estimated flowering (Appendix A), significantly higher functional means were calculated for fields with high compared to low DON levels from four days prior to estimated flowering onwards. When studying the association between T.10T15.CHD at estimated growth stages and DON levels, significant higher values were detected in fields with high compared to low DON levels at BBCH 71, BBCH 81 and BBCH 91 (Appendix A).

The functional means for the daily difference between maximum and minimum observed hourly temperature (T.MINMAXDIFF) were generally stable for both high and low DON levels (Figure 2j,k). High fluctuations of the functional means were observed, but significant differences between the groups were observed in a period around estimated flowering as well as 20 days posterior to estimated flowering, in which 1.5 °C lower values were observed for the group with high DON levels (Figure 2l).

The association between precipitation and DON was assessed by calculating the functional means for the cumulated number of days with precipitation greater than 0.2 mm (P.G02.CD) (Figure 2m,n). The separation between the curves (Figure 2o) showed a consistently higher value of the functional mean for fields with high DON levels, and this difference was significant in the period around sowing and from 20 days prior to estimated flowering onwards. A similar pattern was achieved when starting the accumulation 30 days prior to estimated flowering (Appendix A), with a significant difference from 20 days prior to estimated flowering onwards. When studying the association between P.G02.CD at estimated growth stages and DON levels, significantly higher values were detected in fields with high compared to low DON levels at BBCH 51, BBCH 61, BBCH 71, BBCH 81 and BBCH 91 (Appendix A).

The functional means for daily average relative humidity (RH.A) relative to number of days from estimated flowering had a slightly increasing trend for both the fields with high (Figure 3a) and the fields with low (Figure 3b) DON levels. The separation between the curves (Figure 3c) for the two groups was unstable, but significantly higher levels of relative humidity were observed around estimated flowering for the group with high DON levels. In a five days period around estimated flowering (BBCH 65), the mean relative humidity for the oat fields with high versus low DON levels were 77.9% and 73.3%, respectively. A more distinct separation between the curves was found when comparing the accumulated number of days with relative humidity above 70% (RH.G70.CD) (Appendix A). Here the rate of accumulation differed especially between fields with high versus low DON levels in a 60-day period around estimated flowering; however, significant differences were calculated only at 30 days prior to flowering. When starting the accumulation 30 days prior to estimated flowering (Appendix A), a significantly higher number of days with relative humidity above 70% was calculated for fields with high compared to low DON levels in the period from 10 days prior to estimated flowering onwards.

The functional means for the cumulated number of hours with a vapour pressure deficit lower than 0.635 kPa (VPD.L0635.CHD) were calculated relative to the number of days from estimated flowering (Figure 3d,e). From 80 to 10 days prior to estimated flowering, significantly lower mean values were calculated for fields with high compared to low DON levels, whereas the opposite trend was observed in a short period about 30 days posterior to estimated flowering (Figure 3f). The rate of the accumulation differed especially between fields with high versus low DON levels in the period from 30 days prior to 30 days posterior to estimated flowering. When starting the accumulation 30 days prior to estimated flowering (Appendix A), significantly higher functional means were calculated for fields with high compared to low DON levels from 10 days prior to estimated flowering onwards. When studying the association between VPD.L0635.CHD at estimated growth stages and DON levels, significant lower values were detected in fields with high compared to low DON levels at BBCH 11, BBCH 21 and BBCH 31 (Appendix A).

The functional means for the cumulative number of hours with temperature between 5 and 30 °C combined with a relative humidity greater than 75% (TRH.5T30nRHG75.CHD) were significantly higher for fields with high versus low DON levels from about two weeks posterior to estimated flowering onwards (Appendix A). When considering cumulative hours with temperatures between 15 and 30 °C combined with a relative humidity greater than 80% (TRH.15T30nRHG80.CHD) a similar trend was observed; however, significant differences between the functional means for fields with high compared to low DON levels were only observed between 50 and 30 days prior to estimated flowering (Appendix A). A better association between weather variables and DON levels was detected when the cumulative number of hours with temperature between 9 and 26 °C combined with a relative humidity greater than 75% (TRH.9T26nRHG75.CHD) was considered (Figure 3g,h). The mean functional curves for TRH.9T26nRHG75.CHD were consistently higher for the observations with high DON levels, and the difference was significant in a period around sowing, and from estimated flowering onwards (Figure 3i). The rate of the accumulation differed especially between fields with high versus low DON levels in the period between 25 days prior to and 30 days posterior to estimated flowering. Starting the accumulation 30 days prior to estimated flowering (Appendix A), significantly higher functional means were calculated for fields with high compared to low DON levels already from 10 days prior to estimated flowering onwards. Similar results were observed for TRH.5T30nRHG75.CHD, while only a one-week period prior to estimated flowering was significant for TRH.15T30nRHG80.CHD (Appendix A). When studying the association between TRH.9T26nRHG75.CHD at estimated growth stages and DON levels, significantly lower values were detected in fields with high compared to low DON levels at BBCH 11, whereas significantly higher levels were detected at BBCH 81 and BBCH 91 for fields with high DON levels (Appendix A).

The functional means for the cumulative number of hours with temperature between 5 and 30 °C and vapour pressure deficits lower than 0.635 kPa (TVPD.5T30nVPDL0635.CHD) were calculated relative to the number of days from estimated flowering (Figure 3j–l). Prior to estimated flowering, no significant differences between the groups of fields were observed, whereas three weeks posterior to estimated flowering and onwards significantly higher values were calculated for fields with high compared to low DON levels. Starting the accumulation 30 days prior to estimated flowering (Appendix A), significantly higher functional means were calculated for fields with high compared to low DON levels already from 10 days prior to estimated flowering onwards. Significantly lower values were observed for oat fields with high versus low DON levels when the functional means for the cumulative number of hours with temperature below 5 °C and vapour pressure deficit lower than 0.635 kPa (TVPD.TL5nVPDL0635.CHD) were calculated relative to the number of days from estimated flowering (Appendix A).

The functional means for the cumulative number of hours with temperatures greater than 12 °C and precipitation greater than 0.2 mm TP.TG12nPG02.CHD were significantly lower in fields with high DON levels in a period about 60 to 30 days prior to estimated flowering, whereas from estimated flowering onwards significantly higher functional means were calculated for fields with high compared to low DON levels (Figure 3m–o). The rate of the accumulation differed especially between fields with high versus low DON levels in the periods around estimated flowering and a period prior to harvest. Significantly higher functional means were calculated for fields with high compared to low DON levels from three weeks prior to estimated flowering onwards, when starting the accumulation 30 days prior to estimated flowering (Appendix A). Studying the association between TP.TG12nPG02.CHD at estimated growth stages and DON levels, fields with high DON levels had significantly lower values at BBCH 11 and higher values at BBCH 61, BBCH 71, BBCH 81 and BBCH 91 compared to fields with low DON levels (Appendix A).

## 3. Discussion

To identify weather patterns associated with increased DON levels in oat grain, we conducted a functional data analysis of weather-based time series of data linked to DON content in Norwegian oat grain lots harvested in 201 fields in years 2004–2013. As our dataset did not contain observations of flowering date, which is reported to be the growth stage with highest risk of FHB infection and development of DON in cereal grains, a mathematical model for prediction of growth stages in oats was first developed, validated, and used to estimate the time point of flowering for the oat fields included in the dataset.

Of the 18 phenological models developed in this study, model OPM1.1 was identified to give the best estimate of phenological growth stages of oat varieties in Norway according to our dataset. OPM1.1 gave a more accurate prediction of oat growth stages compared to OPM1.0, a model developed in a previous study [8]. This may be a result of the more adequate dataset used in the current study. For the development of OPM1.1, data on phenological growth stages of oat were recorded throughout the growth season, whereas OPM 1.0 was developed and validated based on growth stage observations mainly recorded at BBCH 59 (heading) and at BBCH 87 (dough development). OPM1.1 gives a more accurate estimate of growth stages than OPM 1.0, especially in the period prior to flowering.

All proposed models were based on the Gompertz growth function, which is a flexible sigmoid curve. The function was previously found to fit well to the phenological oat development in Norway [8], as it is simple and asymmetric. These are both valuable qualities as a model should always be as simple as possible while complexity may lead to overfitting of the data; and secondly, asymmetric since growth is not always symmetrical about the point of inflection.

The proposed models were all based on air temperature and photoperiod, which are the two environmental variables most often considered in crop phenology models [16]. Water stress appears to have a minor effect on the development of wheat and barley [26,27], but was not considered in this study. The effect of precipitation was tested with no effect in Hjelkrem et al. [8].

The OPM1.1 model includes only one weather factor, air temperature. This is in line with other models that also conclude that air temperature is the single most important weather factor for plant development [8,22,24]. The base temperature is the lower threshold temperature for which development stops, and a base temperature of 0 °C was used in our final model. To estimate phenological oat growth stages, a base temperature of both 0 °C [8,24,25] and 5 °C [22,23,28] are frequently used in literature. An increased base temperature was introduced after flowering in a phenological model for winter wheat [29]; however, increasing the base temperature after flowering did not improve our model. The effect of temperature on the developmental rate of wheat was described by cardinal temperatures in Wang and Engel [30], indicating both a temperature range as well as an optimal cardinal temperature for maximum plant growth. However, the use of such a rate function of the temperature response using the cardinal temperatures [31] did not improve our model. The use of minimum and maximum temperatures in a sine curve to approximate the diurnal temperature curve, suggested as an improvement of the ordinal degree-day accumulation [32] was also tested in our dataset; however, with no improvement of the model.

In addition to temperature, photoperiod is considered as an important variable for crop phenology development [16,25,30]. According to Bleken and Skjelvåg [33] and Olesen et al. [22], the effect of photoperiod is strongest before flowering. Still, neither the inclusion of photoperiod during the whole season nor during the pre-flowering period improved the accuracy of our oat phenology model. This might be caused by small variation in photoperiod in the dataset, due to small latitude difference between the locations combined with a smaller time window for sowing (mainly in May).

The oat phenological model was developed based on data of the Belinda variety only, but also validated with several varieties. Oat varieties are known to have different maturing times. According to Peltionen-Sainio and Rajala [28], only the duration of the grain filling period consistently differed between oat varieties from Finland, while the duration of vegetative and generative pre-flowering phases and sub-phases did not. In our study, the different varieties did not contribute much to the variations in model errors, and variety was therefore not included in the model.

The OPM1.1 model predicts the phenological growth stages of oats in Norway based on sowing date and temperature, only. The model was not able to accurately predict the influence of sites and years on oat development. This site-year variation can be an effect of soil type, tillage regimes, measurement errors and environmental factors not included in model development. Also, the plant development is affected by microclimate in the field which might vary highly from the measured air temperature (2 m above ground) at the nearest weather station. Specifically, the model gave the poorest fit to Apelsvoll 2019 and Staur 2019. These are the cases with the earliest and latest sowing date, respectively, (4 May and 4 June) and the effect of the overestimation at Apelsvoll and underestimation at Staur may be due to an effect of this not captured by the model. OPM1.1 is a simple model, but still considered as adequate for predicting the phenological growth stages of Norwegian oats within the normal range of sowing dates.

To develop mathematical models for FHB and DON prediction with adequate accuracy, it is crucial to identify important weather patterns and the time periods where these are of importance for the development of FHB and DON. Functional data analysis was used with smoothing functions, and in order to identify weather patterns of main significance, we presented our data in three different ways: i: Weather patterns reflecting either daily output or cumulative weather variables presented from 90 days prior, to 30 days posterior to estimated flowering, ii: Weather patterns for cumulative time-series of selected weather variables presented from 30 days prior, to 30 days posterior to estimated flowering, iii: Weather patterns for cumulative time-series of selected weather variables presented from sowing according to crop growth resolution instead of on a daily time step. The day of mid flowering was estimated for each field and is used as a base in all the figures. The weather conditions associated with elevated DON levels differed throughout the growth period, as has been reported elsewhere with regards to FHB risk or DON levels in cereals [8,18].

Oat fields with elevated DON levels had a significantly higher relative humidity around estimated flowering compared to fields with low DON levels (77.9 versus 73.3%). This is in agreement with results from a functional study on wheat in which the relative humidity was above 70% in fields with FHB epidemics and 3–4% lower in the remaining fields in the period around flowering [18]. Furthermore, a greater accumulation of hours with humid conditions (high relative humidity or low vapor pressure deficit) was observed around estimated flowering in fields with high versus low DON levels, which corresponds with findings in US wheat [18]. When starting the accumulation at 30 days prior to flowering, a significantly higher number of days with relative humidity above 70% was calculated for fields with high compared to low DON levels in the period from 10 days prior to estimated flowering onwards in our study, indicating that the risk of elevated DON levels in oats increases with pre- and post-flowering moisture. The development of FHB and DON in wheat is observed to increase with moisture periods right after inoculation [34], as well as with extended moisture up to 20 days after flowering [35], which was also indicated in our study on DON levels in oats. Increased development of FHB and mycotoxins may be related to the increased infection rate of *Fusarium* spp. [34], as well as the development and dispersal of *F. graminearum* propagules [15,19,36] with increasing wetness or relative humidity. When starting the accumulation at 30 days prior to flowering, a significantly higher number of accumulated hours was detected from about two weeks posterior to estimated flowering onwards for many of the variables included in our study (VPD.L0635.CHD, TVPD.5T30nVPDL0635.CHD, TRH.5T30nRHG75.CHD and TRH.9T26nRHG75.CHD), indicating that these weather conditions during flowering may increase the risk for fungal infection and DON accumulation in oats. Wet or humid weather at flowering is found to favour *Fusarium* infection and development of FHB [17,37]. In line with our results, a higher accumulated number of hours with temperature between 5 and 30 °C combined with a relative humidity above 75% was observed at flowering for FHB epidemics versus non-epidemics in US wheat [18]. When we increased the minimum temperature (TRH.15T30nRHG80.CHD), the number of accumulated hours from 30 days prior to flowering was significant higher in fields with high versus low DON levels in a 10-day window prior to flowering only. Comparable weather conditions have been identified to favour *F. graminearum* perithecia formation [36] and discharge of ascospores [38]. The association between humidity and FHB development is underlined in a study of wheat in which the average relative humidity around wheat flowering was included in 14 out of 15 FHB models [11]. In our study, as well as in the abovementioned studies, humid conditions around flowering seem to be associated with increased FHB and DON levels in harvested grain. Thus, weather factors that incorporate relative humidity should be considered in the development of FHB prediction models. The positive association between DON levels and humid conditions was evident way beyond the flowering period in our study.

In contrast to the weather conditions at flowering, humid weather in the period from sowing until stem elongation was associated with reduced DON levels. Oat fields with elevated DON levels had less humid conditions (a significantly lower number of hours with a vapor pressure deficit below 0.635 kPa) compared to fields with low DON levels. This difference was also evident when including temperatures below 5 °C only (TVPD.TL5nVPDL0635.CHD), but not for temperatures between 5 and 30 °C. This indicates that cool and moist weather in the early growth periods is associated with low DON levels. This may be related to the reduced production of perithecia [36], as well as to a reduced dispersal of *F. graminearum* ascospores at low temperatures despite saturation humidity [19]. We also speculate whether cool and moist weather may stimulate competing saprophytes and result in a reduction of the fields’ *Fusarium*-inoculum level. Additionally, cool and moist conditions around sowing may indicate that these fields were sown early in the growth season (relative to fields with high DON levels) which may reduce the risk of receiving *Fusarium*-inoculum at mid flowering from surrounding fields. We surmise that cool and humid weather in the period around sowing was associated with reduced DON levels.

Oat fields with elevated DON levels had a lower difference between minimum and maximum daily temperatures at estimated flowering. Shah et al. [18] also observed less variable temperatures around wheat flowering during FHB epidemics versus non-epidemics. According to Dai et al. [39], the daily difference in minimum and maximum temperature is negatively correlated with cloud cover and accompanying moisture. This relationship was confirmed in our dataset in which the average daily relative humidity was above 70% for most of the fields with a daily difference in temperature below 4 °C. The figures showing a difference between the functional mean for daily average relative humidity and for the daily difference between minimum and maximum air temperature in our study nearly mirror each other, indicating that periods with low differences in temperature correspond with high relative humidity. Temperature data may be more accessible than data on relative humidity. However, the use of daily differences in temperature could only be a surrogate method to identify periods with high relative humidity when direct measurements are not available.

Oat fields with elevated DON levels had a higher number of accumulated days with rain throughout the whole growth period. When starting the accumulation 30 days prior to flowering, a significantly higher number of rainy days was observed in fields with high versus low DON levels in the period from three weeks prior to estimated flowering onwards. When starting the accumulation from sowing, a significantly higher number of rainy days was observed in fields with high versus low DON levels from heading onwards, indicating that rainy conditions around flowering increases the DON risk in oats. This corresponds with observations in *F. graminearum* inoculated wheat, where increased DON accumulation was connected to wet periods extended until about three weeks after flowering [35], as well as to post flowering moisture under controlled environmental conditions [40]. As mentioned earlier, this may be related to increased infection rate of *Fusarium* spp., as well as development and dispersal of *F. graminearum* propagules with increasing wetness. Besides, both macroconidia and ascospores of *F. graminearum* may be splash-dispersed [41]. Thus, rain may increase the inoculum pressure of *F. graminearum*. When starting the accumulation 30 days prior to flowering, oat fields with elevated DON levels had a significantly higher number of hours with rain and temperatures above 12 °C compared to field with low DON levels in the period from three weeks prior to estimated flowering onwards, which may be related to a reduced development of *F. graminearum* perithecia at low temperatures [36]. Shah et al. [18] did not include rainfall in their functional data analysis of weather variables linked to FHB epidemics in wheat as they considered this variable to be more site-specific than the resolution of their input data. In our dataset, the weather stations are located at a maximum 15 km distance from the fields, which we consider as being within an acceptable distance. In several studies, high FHB or DON levels have been associated with rainy conditions around flowering. However, the occurrence of rain can be very site specific, and thus it may be difficult to achieve a relevant resolution of data.

In our dataset, oat fields with elevated DON levels had lower daily average temperatures from around estimated flowering onwards, compared to fields with low DON levels. Likewise, significantly lower temperatures during flowering were observed for FHB epidemics vs. non-epidemics in US wheat [18]. The daily average temperatures at flowering were between 15 and 20 °C in both studies. A negative association between elevated DON levels and warm weather at flowering was also reported by Klem et al. [9]. Fields with high DON levels had also a higher number of hours with temperature between 10 and 15 °C compared to fields with low DON levels. These conditions were mainly below the average observed temperature for the time period in our dataset and represent conditions that are sub optimal for production and maturation of *F. graminearum* perithecia [36,42]. Higher DON content in fields with low daily average temperatures could perhaps be related to increased survival of *F. graminearum* ascospores after exposure to moderate, compared to warm, temperatures both in dry [19,20] or humid conditions [43]. Lower temperatures will also slow down the phenological development of the oat crop, thereby increasing the duration of the flowering period, thus increasing the window in which the plants are especially susceptible to *Fusarium* infection. The output from our functional data analysis of weather conditions of 10–15 °C combined with vapour pressure deficits below 0.635 kPa resembled the one with temperature only, indicating relatively humid conditions at these temperatures. In line, a reduction in temperatures was associated with a reduction in vapor pressure deficits (increased humidity) in our dataset, conditions which may additionally increase the survival of *F. graminearum* ascospores [15], as well as the development of FHB. Temperature related weather variables in the period around flowering are often associated with FHB epidemics or DON accumulation [5,8,11,12,18]. However, these variables often include a temperature range exceeding the one discussed here.

In contrast to the weather conditions at flowering, lower daily average temperatures in the period from sowing until stem elongation were associated with reduced DON levels. Oat fields with elevated DON levels had generally higher daily average temperatures compared to fields with moderate to low DON levels. A higher number of hours with temperatures between 10 and 15 °C during this period was also identified for fields with high versus low DON levels. These temperature conditions can be regarded as above average for the time period, as the average temperature in a five-day window after sowing in our dataset was 10.4 °C. In our previous DON-model, similar weather conditions during germination/seedling growth were also associated with increased DON risk in oats [8]. However, when starting the accumulation from the actual sowing date, there was no difference in the number of accumulated hours between 10 and 15 °C for the group of fields with high versus low DON levels in the period after sowing. Thus, the higher daily average temperatures in fields with high DON levels may be explained by fewer hourly incidences of temperatures below 9 °C. Similar associations between elevated DON levels and warm weather during early phenological growth stages have been recorded in wheat [9,10]. An association between temperatures below 9 °C and low DON levels may be related to a suppression of the development and maturation of *F. graminearum* perithecia on crop residues at low temperatures [36,42,44]. We surmise that elevated DON or FHB levels may be related to increased daily average temperatures in the period around sowing.

Correlation is expected between several of the weather variables included in our dataset. For example, the weather variables that include rain, relative humidity, or vapor pressure deficit, all relate to the amount of water vapor in the air. Likewise, low temperatures are often associated with rainy conditions, whereas high temperatures are often associated with dry and sunny weather. A correlation was shown between several of the variables in our dataset, and the magnitude of the correlation differed between the time periods in which the data were collected. For example, the temperature range was narrower during the period of flowering compared with the whole growth season. Consequently, the output of two moisture related variables, as VPD.L0635.CHD and TRH.9T26nRHG75.CHD, was more highly correlated (r = 0.91) when collecting data within a narrow time interval around flowering, than during the entire growing season (r = 0.74). The inclusion of highly correlating variables in a model should be avoided because of overfitting problems. The selection of parameters during model development may be a result of the magnitude of correlation or simply of available data and was not the purpose of this study.

The functional study indicates that DON contamination in harvested grain is associated with weather conditions during the whole growth period of oats, from sowing to harvest. However, the largest difference in the rate of accumulation (hours, days) of specific weather conditions for oat fields with high versus low DON levels was observed in the period around flowering. In line with many studies in cereals, this indicates that weather conditions during flowering are especially important for the infection with *Fusarium* spp. and the accumulation of DON. To get an overview of the general weather pattern and whether this may be associated with DON content in harvested grain, the figures with time spans from 90 days prior to 30 days posterior to estimated flowering may be best suited. To identify weather patterns of significance in the time period around sowing, functions derived according to crop growth resolution may be best suited. Time series presented from 30 days prior to 30 days posterior to estimated flowering may be best suited to identify weather patterns of significance in the time period around flowering. For many of the weather variables, the association with DON levels in harvested oats differed throughout the growth period e.g., in the beginning of the growth season VPD.L0636.CHD was negatively associated with increased DON levels; however, this trend was opposite in the period around flowering onwards. Some weather conditions indicated the same trend throughout the whole growth season, such as T.10T15.CHD, which may indicate these as strong candidates to be included in prediction models.

## 4. Materials and Methods

### 4.1. Field Trial Observations Included in the Oat Phenology Model

To develop the oat phenology model, thorough observations were performed in trials for the assessment of the Value for Cultivation and Use (VCU) of oat varieties in years 2015 to 2019. The VCU trials were situated at four different locations in the main cereal cultivating district in south-eastern Norway, totalling in 11 combinations of site and year (Appendix A). The following data were included in the model development: sowing date, field location, and weekly observations of phenological growth stages determined according to the BBCH scale [45] for selected oat varieties throughout the growth season. All fields were located within a maximum distance of 10 km from a weather station, and two of the fields shared the same weather station as the closest. A total of 13 different oat varieties were included in the model, of which six varieties (Belinda, Ringsaker, Haga, Odal, Vinger and Våler) were grown at all site-year combinations. A total of 1089 records of BBCH were collected in the fields between sowing and harvest, covering the growth stages BBCH 11 to BBCH 94. Most of the records (62%) were on the late maturing varieties (Odal, Belinda, Vinger, Hurum, Våler and Årnes), while the remaining records (38%) concerned the early maturing varieties (Ringsaker, Haga, Gimse, Dovre, Akseli, Hurdal and Avetron). The late maturing variety Belinda was used as a reference-variety for model development in this study, with 123 field observations between BBCH 11 and BBCH 94.

### 4.2. Oat Fields Included in the Functional Data Analysis

Representative oat grain samples were collected at farms or at delivery points of oats from farmers’ fields in the main cereal district in Norway (south-east) from 2004 to 2013. Sowing dates and the nearest weather stations were recorded for each field. The dataset is identical to the dataset used in Hjelkrem et al. [8], except that the following samples were removed: samples of the variety Bessin, which is highly susceptible to *F. graminearum* [3], samples harvested from fields with a distance to the nearest weather station of more than 15 km, and samples harvested from fields located in mid-Norway. The final dataset consisted of oat samples from 201 fields with the majority from 2006–2008 and 2012. The oat grain samples were processed and analysed for DON as described in Hjelkrem et al. [8].

### 4.3. Weather and Environmental Data

Weather records were provided by Agrometeorology Norway (https://lmt.nibio.no/, accessed on 17 November 2021), with air temperature (T; °C), precipitation (P; mm) and relative humidity (RH; %) measured at an hourly temporal resolution.

Data on relative humidity are often included in disease risk models [4,8,40,42]. However, humidity sensors are sensitive for the erosion of calibration [46] and a drift in measurements is common over time [47]. Such a drift was observed for relative humidity measurements from a selection of the weather stations included in our study, with a clear drift when considering the maximum hourly relative humidity value for each month through the seasons (Appendix A). However, the minimum output values had a constant pattern across the years (Appendix A). By comparing the data output from one sensor with a drift in measurements and one without, with both placed in the exact same spot, we were able to identify measuring ranges that were especially exposed to drift. Inequalities in relative humidity output from the two sensors predominantly occurred for measurements above 70% (Appendix A). In line with Eccel [47] we developed an algorithm for a linear correction of relative humidity readings above 70%, ranging from 0% (at measures equal to 70%) to the value necessary to reach 100% (at the highest measure actually recorded by the instrument) (Appendix A). As the sensors operate with an accuracy of ±3% for relative humidity above 90% (Agrometeorology Norway), the corrections of output data were performed for weather stations from which the maximum hourly measured relative humidity throughout a year was below 97%.

The vapour pressure deficit (vpd; kPa) was calculated as the difference between the saturated vapour pressure [48] and the actual vapour pressure [49]. Photoperiod (λ; h), which is the number of daylight hours, was estimated daily from the latitude of the nearest weather station and the day of the year [50].

To develop an improved phenological model for estimating growth stages in oat, daily air temperature records and the estimated photoperiod were tested as possible variables.

To identify the possible relationship between DON contamination in oats and weather conditions, a total number of 14 weather variables were calculated based on weather records and further evaluated by using functional data analysis (Table 2). The weather variables were selected based on associations between weather conditions and development of *F. graminearum*, FHB and/or DON in cereals (mainly wheat) reported in previous studies [4,8,11,12,13,14,17,18,36,38,42,44,51]. Possible associations between selected weather variables were also examined.

### 4.4. Development of an Improved Model for Prediction of Phenological Growth Stages of Oats

In order to predict the phenological growth stages of Norwegian oats, we developed and validated eighteen different mathematical models (Appendix A). The plant development rate for most cereal crops, are sigmoid functions of degree-days [16]. Accordingly, the flexible asymmetric sigmoid Gompertz growth function [52] was used as a base function for all proposed models in this study, with different adjustments of degree-day accumulation (Equation (1)).
(1)$$GS_{i}=GS_{max}\cdot e^{-\alpha \cdot e^{-\beta \cdot {\mathrm{DD}}_{i}}}$$

Here *GS_i_* is the predicted phenological growth stage according to BBCH at day *i*, *GS_max_* is the upper asymptote, *α* sets the displacement along the *x*-axis, *β* sets the growth rate and *DD_i_* is the adjusted degree-day accumulation from sowing to day *i*. The models depended on air temperature alone or in combination with the photoperiod, which are the variables most often considered in crop phenology models [16].

Calibration is the process of finding the best parameter estimates for the different models, based on field data. In line with Hjelkrem et al. [8], the Bayesian framework was used with the Markov chain Monte Carlo (MCMC) algorithm Random walk Metropolis [53]. Markov chains of the posterior parameter distributions were generated based on wide uniform prior parameter distributions combined with new incorporated information through the data, reflected by the likelihood function [54]. Point estimates were calculated from the Markov chains, as maximum a posteriori probability (MAP) estimates for the unknown model parameters.

The models were developed and validated through the cross site-year validation technique [55], to use the limited number of available field data in an optimal way. The root mean square error of validation (RMSEV) was estimated for each split in the cross site-year validation and given as the mean value and standard deviation over the 11 splits. The model with the lowest RMSEV was regarded as the best model. With models of an approximately similar error term, the simplest model (with fewest unknown parameters) was selected as the overall best.

In order to investigate to what extent a variety-specific growth model was required, the models were developed by using data from the variety Belinda, only. The models were then validated on all varieties included in the field trials in this study, both separately and merged.

### 4.5. Functional Data Analysis of Weather Conditions Related to DON in Oat Grain

Functional data analysis encompasses the methodology of time series data and was used to identify weather conditions of importance for development of DON in oats. The weather variables were selected based on previous findings on the relationship between weather and the development of FHB in cereals (Table 2). A smooth functional curve was estimated for each weather variable (Equation (2)).
(2)$$y_{ij}=x_{i}\left(t_{ij}\right)+\epsilon_{ij}$$
where *y_ij_* denotes the measured time series of weather variables. The time series are processed on a daily time step *j* for a replication *i* of combinations of site, year and starting day. These values are further converted into functions *x_i_*(*t_ij_*)*,* that represent a smooth curve for each replicate. When no errors ($\epsilon_{ij}$) are allowed, this is called interpolation, while it is denoted as smoothing otherwise [56]. We conducted smoothing with cubic B-spline (basis spline) functions, where a cubic spline was constructed from third-order polynomial segments. The day of mid flowering (BBCH 65) was estimated for each field by using the phenological model (OPM1.1), and the smoothing functions derived from 90 days prior to 30 days posterior to estimated flowering. For some cumulative time-series, the smoothing functions were also derived from 30 days prior to 30 days posterior to estimated flowering. Additionally, the day of specific phenological growth stages (BBCH 11, BBCH 21, BBCH 31, BBCH 41, BBCH 51, BBCH 61, BBCH 71, BBCH 81 and BBCH 91) was estimated from OPM1.1, with smoothing functions derived according to the crop growth resolution instead of on a daily time step.

The curves were separated into two pools, one pool associated with low DON levels (<500 µg DON per kg grain) and one associated with high DON levels (≥500 µg DON per kg grain). The pointwise functional mean was calculated for each pool according to Equation (3).
(3)$$\bar{x}\left(t_{j}\right)=N^{-1}\overset{N}{\underset{i=1}{\sum }}x_{i}\left(t_{ij}\right)$$

The difference between the curves associated with low and high DON levels were also calculated, and the associated significance (5% level) was estimated through a permutation test, run with 1000 permutations.

### 4.6. Software

All calculations and modelling were done using Matlab R2020b. For functional data analysis, the cubic smoothing spline (csaps) in the Curve Fitting Toolbox was used, while the Permutation Test used to denote significance was calculated in line with Krol [57].

## 5. Conclusions

We conclude that functional data analysis can be a useful tool to reveal weather patterns associated with DON levels in oats. In our dataset, the DON content of harvested oat grain was especially associated with weather conditions in the period around flowering, in line with similar analysis presented for FHB epidemics and DON contamination in wheat grain. Indeed, many of the weather variables identified to be associated with FHB or DON levels in wheat in previous studies were associated with DON levels in oats in our study. Oat fields with elevated DON levels had generally higher daily average temperatures and fewer days with cool temperatures in the period around sowing, less humid conditions at early growth stages, and lower daily average temperatures as well as more rain or humid conditions combined with moderately warm temperatures in a period prior to flowering onwards, compared to fields with moderate to low DON levels. Our findings can be used as a basis for the development of mathematical models to predict DON risk in oats, both as part of a decision support system for farmers to assist in decisions on the need, or not, of fungicide application to control FHB in a particular field as well as for predicting the quality at harvest. The phenological model we have developed can be used to estimate the time point of specific phenological growth stages from sowing to ripening of Nordic spring oat varieties, by using sowing date and air temperature in the specific field as the only input variables.

## Figures and Tables

**Figure 1 plants-11-00073-f001:**
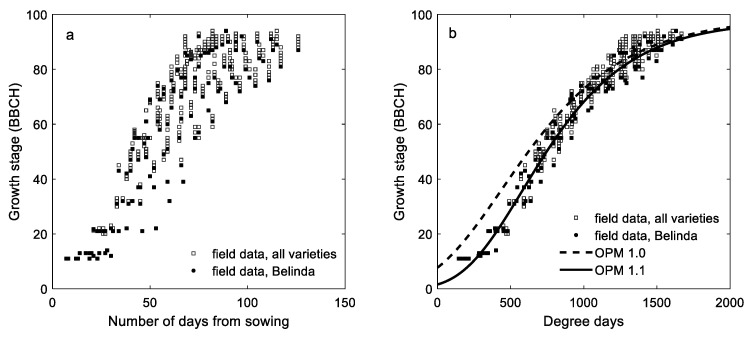
Observed growth stage (BBCH) data for six varieties versus (**a**) number of days from sowing and (**b**) degree days, including output data from the oat phenology model OPM 1.0 [8] and OPM 1.1 (Model 1 derived in this study). Observations on variety Belinda are marked in black.

**Figure 2 plants-11-00073-f002:**
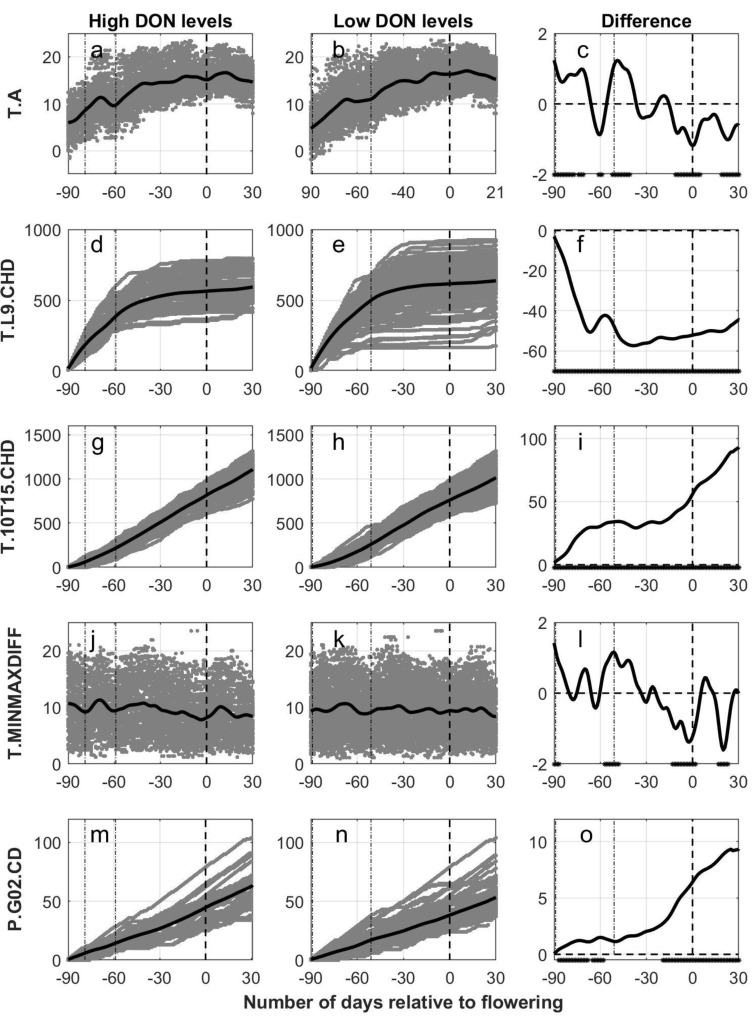
Calculated output data (gray) for five weather variables versus the number of days relative to estimated flowering in oats, grouped according to levels of deoxynivalenol (High DON levels, first column of panels (subfigures (**a**,**d**,**g**,**j**,**m**)): DON ≥ 500 µg pr kg harvested oat. Low DON levels, second column of panels (subfigures (**b**,**e**,**h**,**k**,**n**)): DON < 500 µg per kg harvested oat). Black lines indicate the functional mean values of the smooth functions derived from individual sets of weather data. In the third column of panels (subfigures (**c**,**f**,**i**,**l**,**o**)), the difference between the functional mean curves (high–low) are given as black lines with a dot at the bottom of each panel indicating significant differences between the mean values (5% level). Weather data is presented from 90 days prior to 30 days posterior to flowering (dashed line at 0) estimated with the use of the phenological model (OPM1.1). In all panels, the two dash-dotted lines indicate the start and end of the period of sowing for the different fields. T.A = daily average temperature, T.L9.CHD = cumulative number of hours with air temperature lower than 9 °C, T.10T15.CHD = cumulative number of hours with air temperature between 10 and 15 °C, T.MINMAXDIFF = daily difference between the observed maximum and minimum observed hourly temperature and P.G02.CD = cumulative days with precipitation greater than 0.2 mm.

**Figure 3 plants-11-00073-f003:**
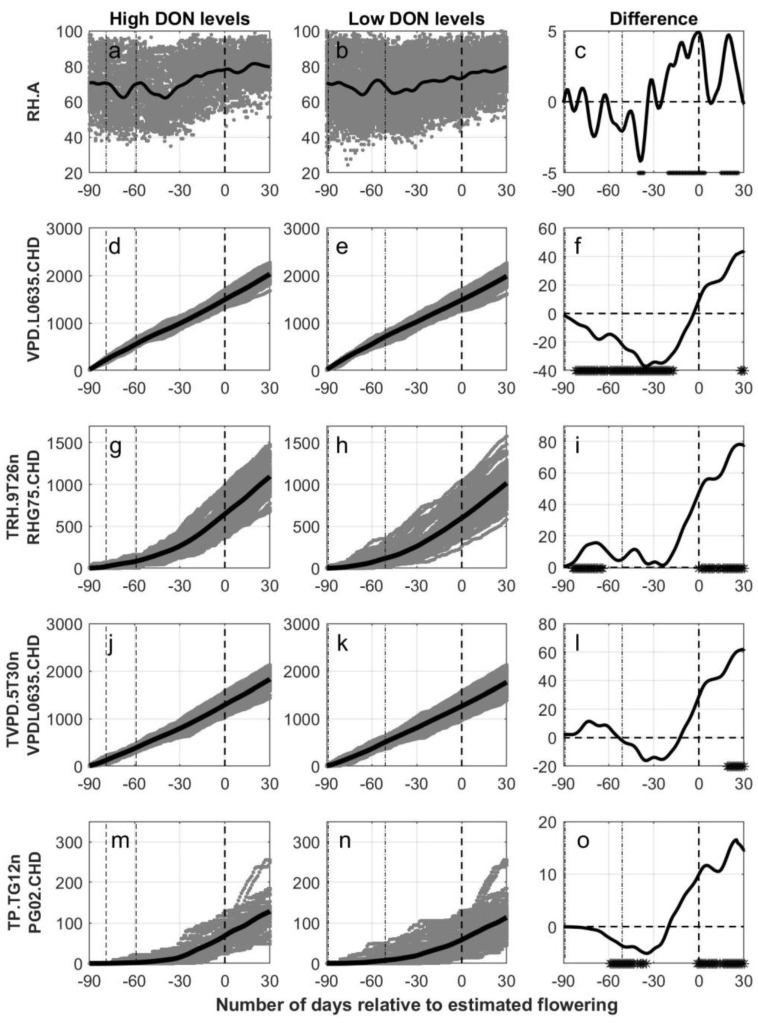
Calculated output data (gray) for five weather variables versus the number of days relative to estimated flowering in oats, grouped according to levels of deoxynivalenol (High DON levels, first column of panels (subfigures (**a**,**d**,**g**,**j**,**m**)): DON ≥ 500 µg pr kg harvested oat. Low DON levels, second column of panels (subfigures (**b**,**e**,**h**,**k**,**n**)): DON < 500 µg per kg harvested oat). Black lines indicate the functional mean values of the smooth functions derived from individual sets of weather data. In the third column of panels (subfigures (**c**,**f**,**i**,**l**,**o**)), the difference between the functional mean curves (high–low) are given as black lines with a dot at the bottom of each panel indicating significant differences between the mean values (5% level). Weather data is presented from 90 days prior to 30 days posterior to flowering (dashed line at 0) estimated with the use of the phenological model (OPM1.1). In all panels, the two dash-dotted lines indicate the start and end of the period of sowing for the different fields. RH.A = daily average relative humidity, VPD.L0635.CHD = cumulative number of hours with vapour pressure deficit lower than 0.635 kPa, TRH.9T26nRHG75.CHD = cumulative number of hours with temperature between 9 and 26 °C combined with a relative humidity greater than 75%, TVPD.5T30nVPDL0635.CHD = cumulative number of hours with temperature between 5 and 30 °C combined with vapour pressure deficit lower than 0.635 kPa and TP.TG12nPG02.CHD = cumulative number of hours with temperature above 12 °C combined with precipitation greater than 0.2 mm.

**Table 1 plants-11-00073-t001:** Estimated root mean square error of validation (RMSEV) of the different models fit to validation data from the varieties Belinda, Ringsaker, Haga, Odal, Vinger and Våler separately and for all 13 varieties included in this study merged given as mean value (and standard deviation) over the splits in the cross site-year validation.

	Belinda	Ringsaker	Haga	Odal	Vinger	Våler	All Varieties
Model 1	4.2 (1.0) ^2^	4.3 (1.7) ^1^	4.2 (1.5) ^2^	4.1 (1.5) ^1^	4.3 (1.3) ^1^	4.1 (1.3) ^1^	4.3 (1.3) ^1^
Model 2	4.2 (1.0) ^2^	4.4 (1.7) ^2^	4.2 (1.5) ^2^	4.2 (1.4) ^2^	4.3 (1.3) ^1^	4.1 (1.3) ^1^	4.4 (1.3) ^2^
Model 3	5.2 (2.5)	5.6 (2.7)	5.4 (2.5)	5.1 (2.5)	5.4 (2.6)	5.3 (2.4)	5.4 (2.6)
Model 4	4.3 (0.9)	4.6 (1.5)	4.4 (1.3)	4.3 (1.2)	4.5 (1.1)	4.3 (1.1)	4.5 (1.1)
Model 5	5.4 (1.0)	6.1 (1.3)	5.8 (1.1)	5.6 (1.1)	5.8 (1.2)	5.5 (1.1)	5.9 (1.1)
Model 6	4.3 (0.9)	4.5 (1.5)	4.3 (1.3)	4.2 (1.2) ^2^	4.5 (1.1)	4.3 (1.1)	4.5 (1.1)
Model 7	4.3 (1.0)	4.6 (1.5)	4.4 (1.3)	4.3 (1.3)	4.5 (1.2)	4.3 (1.2)	4.5 (1.2)
Model 8	4.4 (1.2)	4.6 (1.7)	4.5 (1.6)	4.5 (1.6)	4.5 (1.5)	4.4 (1.4)	4.6 (1.4)
Model 9	4.6 (1.1)	4.8 (1.7)	4.7 (1.5)	4.6 (1.4)	4.8 (1.3)	4.6 (1.3)	4.8 (1.4)
Model 10	4.3 (1.1)	4.5 (1.8)	4.3 (1.6)	4.3 (1.5)	4.4 (1.4) ^2^	4.2 (1.4) ^2^	4.4 (1.4) ^2^
Model 11	5.3 (2.4)	5.6 (2.7)	5.4 (2.5)	5.2 (2.4)	5.4 (2.6)	5.4 (2.4)	5.5 (2.6)
Model 12	4.4 (1.2)	4.6 (1.9)	4.4 (1.7)	4.4 (1.6)	4.6 (1.5)	4.4 (1.6)	4.6 (1.5)
Model 13	4.3 (1.0)	4.5 (1.6)	4.3 (1.4)	4.3 (1.4)	4.4 (1.2) ^2^	4.3 (1.3)	4.4 (1.3) ^2^
Model 14	4.3 (1.0)	4.5 (1.6)	4.3 (1.4)	4.3 (1.4)	4.5 (1.3)	4.3 (1.3)	4.5 (1.3)
Model 15	4.4 (2.0)	4.9 (2.5)	4.6 (2.3)	4.6 (2.2)	4.6 (2.2)	4.4 (2.2)	4.7 (2.2)
Model 16	6.5 (3.6)	7.3 (4.0)	6.9 (3.8)	6.6 (3.6)	6.8 (3.8)	6.7 (3.7)	7.0 (3.8)
Model 17	4.2 (1.0) ^2^	4.3 (1.5) ^1^	4.1 (1.4) ^1^	4.2 (1.5) ^2^	4.3 (1.3) ^1^	4.1 (1.4) ^1^	4.3 (1.2) ^1^
Model 18	4.1 (1.1) ^1^	4.4 (1.5) ^2^	4.1 (1.4) ^1^	4.2 (1.4) ^2^	4.3 (1.2) ^1^	4.1 (1.3) ^1^	4.3 (1.2) ^1^

^1^ The model with lowest RMSEV. ^2^ The model with the second lowest RMSEV.

**Table 2 plants-11-00073-t002:** Weather variables selected for use in the functional data analysis.

Abbreviation ^1^	Description (Associated With) ^2^
T.A	Daily average temperature
T.L9.CHD	Cumulative number of hours with temperature lower than 9 °C
T.10T15.CHD	Cumulative number of hours with temperature between 10 and 15 °C
T.MINMAXDIFF	Daily difference between the maximum and minimum hourly temperature
P.G02.CD	Cumulative number of days with precipitation greater than 0.2 mm
RH.A	Daily average relative humidity
RH.G70.CD	Cumulative number of days with daily relative humidity above 70%
VPD.L0635.CHD	Cumulative number of hours with vapor pressure deficit lower than 0.635 kPa
TRH.5T30nRHG75.CHD	Cumulative number of hours with temperature between 5 and 30 °C combined with relative humidity greater than 75%
TRH15T30nRHG80CHD	Cumulative number of hours with temperature between 15 and 30 °C combined with relative humidity greater than 80%
TRH.9T26nRHG75.CHD	Cumulative number of hours with temperature between 9 and 26 °C combined with relative humidity greater than 75%
TVPD.5T30nVPDL0635.CHD	Cumulative number of hours with temperature between 5 and 30 °C combined with vapor pressure deficit lower than 0.635 kPa
TP.TG12nPG02.CHD	Cumulative number of hours with temperature greater or equal to 12 °C combined with precipitation greater than 0.2 mm
TVPD.TL5nVPDL0635.CHD	Cumulative number of hours with temperature lower than 5 °C combined with vapor pressure deficit lower than 0.635 kPa

^1^ The abbreviation of various weather variables used in this manuscript. ^2^ Observations of disease development, grain quality, or various parts of the life cycle of *Fusarium graminearum* reported to be associated with this weather variable.

## Data Availability

All data used in and created by this study are included in this publication as tables, figures, and Appendix A.

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
