# Peer review of "Weather Patterns Associated with DON Levels in Norwegian Spring Oat Grain: A Functional Data Approach"

_plants, 2021, doi:10.3390/plants11010073_

Round 1

Reviewer 1 Report

In this manuscript the authors report on the development of an improved model for prediction of phenological growth stages of oats, thorough observations performed in trials 650 for the assessment of the Value for Cultivation and Use (VCU) of oat varieties in years 2015 to 2019 in south-eastern Norway, and collection of weather data. This model improved a previous one. It was used for defining the phenological stages in a functional data analysis of weather conditions related to DON (a mycotoxin) concentration in oat grain. Actually, it was new analyses of a dataset previously used by the authors in a paper published in 2017.  Weather and environmental data were collected from weather stations as close as possible to the fields.

Such functional data analysis proved to be a useful tool to reveal weather patterns associated with DON levels in oats, and the findings of this work can be used as a basis for the development of mathematical models to predict DON risk in oats.

I have no specific comment for improving this interesting manuscript.

Author Response

Reviewer #1:

Such functional data analysis proved to be a useful tool to reveal weather patterns associated with DON levels in oats, and the findings of this work can be used as a basis for the development of mathematical models to predict DON risk in oats.

I have no specific comment for improving this interesting manuscript.

Reply: Thank you so much for positive feedback on our manuscript!

Reviewer 2 Report

The manuscript is very interesting. The statistical functional analysis has been correctly applied. Rich graphical analysis. There are no critical remarks. 

Author Response

Reviewer #2:

The manuscript is very interesting. The statistical functional analysis has been correctly applied. Rich graphical analysis. There are no critical remarks.

Reply: Thank you so much for positive feedback on our manuscript!

Reviewer 3 Report

Generally, I like the paper, and my opinion is that it represents a significant advancement of the field. The figures are of poor quality and the discussion section is lengthy.

I only have several minor comments, as the paper is very well written and polished.

L80 is there a reference to some whitepaper of the service?

L107 there should be a reference to models in table S3

L110 in my opinion, models are not well specified. There should be at least one supplementary table with model details.

L148 I like the inclination of authors towards the parsimonious solution.

L148-155 is it valid to address the model improvement as an aim of the study, only to re-calculate the model parameters in MCMC simulations of the existing, well performing baseline model?

L191 in my opinion, Figure 2 should be moved to supplementary data as it does not show anything relevant for the discussed topic, but rather the analysis of known properties of VPD, temperature and humidity.

Author Response

Reviewer #3:

Generally, I like the paper, and my opinion is that it represents a significant advancement of the field. The figures are of poor quality and the discussion section is lengthy.

I only have several minor comments, as the paper is very well written and polished.

Reply: Thank you so much for positive feedback on our manuscript! New figures of better quality are included. We agree that the discussion part is long but struggle to shorten it down. Therefore, we have not made any further changes in the discussion in this round.

L80 is there a reference to some whitepaper of the service?

Reply: Unfortunately, there are no whitepaper to refer to according to VIPS, only the webpage.

L107 there should be a reference to models in table S3

Reply: We agree, and a reference to Table S3 is given in L102 and in L108. In this new version, Table S3 is denoted Table S1.

L110 in my opinion, models are not well specified. There should be at least one supplementary table with model details.

Reply: We agree. Table S3 (in this new version denoted Table S1) is now increased by including more details about the models.

L148 I like the inclination of authors towards the parsimonious solution.

Reply: Thank you.

L148-155 is it valid to address the model improvement as an aim of the study, only to re-calculate the model parameters in MCMC simulations of the existing, well performing baseline model?

Reply: We agree that the model improvement is only a re-calibration of model parameters. The aim was still to improve the model, and 18 options were tested, although the new model turned out to be just a re-calibration of the old model. In-line with comments, we have added “, with a new structure or at least more finely calibrated,” in L91-92. In L96 improved is replaced with updated. 

L191 in my opinion, Figure 2 should be moved to supplementary data as it does not show anything relevant for the discussed topic, but rather the analysis of known properties of VPD, temperature and humidity.

Reply: We agree, and Figure 2 is now moved to the supporting information as Figure S3.